# Management of Needle-Eating Caterpillars Associated with *Pinus massoniana* and *P. merkusii* in Vietnam

Dao Ngoc Quang [1,*], Pham Quang Thu [1], Nguyen Minh Chi [1], Le Van Binh [1], Nguyen Quoc Thong [1], Nguyen Hoai Thu [1], Vo Dai Nguyen [2] and Bernard Dell [1,3]

[1] Forest Protection Research Centre, Vietnamese Academy of Forest Sciences, Duc Thang, Bac Tu Liem, Hanoi 11910, Vietnam; phamquangthu@vafs.gov.vn (P.Q.T.); nguyenminhchi@vafs.gov.vn (N.M.C.); levanbinh@vafs.gov.vn (L.V.B.); thongttbvr@gmail.com (N.Q.T.); hoaithu.fsiv@gmail.com (N.H.T.); B.Dell@murdoch.edu.au (B.D.)
[2] Silviculture Research Institute, Vietnamese Academy of Forest Sciences, Duc Thang, Bac Tu Liem, Hanoi 11910, Vietnam; vodainguyenvnuf@gmail.com
[3] Agriculture and Forest Sciences, Murdoch University, Murdoch 6150, Australia
\* Correspondence: daongocquang@vafs.gov.vn; Tel.: +84-913-570-077

**Abstract:** Pine tussock moth (*Dasychira axutha* Collenette (Lymantriidae)) and masson pine caterpillar (*Dendrolimus punctatus* Walker (Lasiocampidae)) cause serious damage to *Pinus massoniana* and *P. merkusii* plantations in Vietnam. An integrated pest management (IPM) program is required to control these pests. Therefore, this study was undertaken to identify damage thresholds and to evaluate control options for implementing IPM in the Northeast and North Central regions of Vietnam. Three damage thresholds were considered: <25%, 25%–50% and >50% loss of leaf area. Control options were manual collection of eggs and pupae and the use of light traps and biological agents. An initial IPM model was developed for each pest and these gave control efficacies of 82.4% (*Da. axutha* on *P. massoniana*) and 77.8% (*De. punctatus* on *P. merkusii*). Six extended IPM models were undertaken by forest companies giving pest control efficacies of 79.2%–85.9%. The collection of pine resin provides an important source of income to local communities and the resin yield in the IPM models increased by 20.9%–22.7% compared to untreated control plots. The IPM protocols would help in developing policies that promote a more sustainable management of forest pests in Vietnam, thus reducing dependence on the use of insecticides of environmental and public health concerns.

**Keywords:** damage threshold; *Dasychira axutha*; *Dendrolimus punctatus*; integrated pest management; masson pine caterpillar; pine tussock moth; plantation forestry; resin yield

## 1. Introduction

*Pinus* is the largest genus in the Pinaceae family, with 111 species, mainly naturally distributed in temperate parts of the northern hemisphere [1,2]. A small number of species occurs in tropical and sub-tropical climate zones, including *P. massoniana*, native to Taiwan, a wide area of central and southern China and northern Vietnam [2,3], and *P. merkusii*, which has a disjunct distribution in Southeast Asia, including northern Sumatra and the Philippines [2,4,5]. Both species have been used for large-scale afforestation [3,4,6,7]. The total forest area of pine plantations in Vietnam is about 300,000 hectares, comprising, mainly, *Pinus massoniana* (160,000 ha) and *P. merkusii* (90,000 ha) [8–10].

Two species of Lepidoptera, *Dasychira axutha* Collenette (Lymantriidae) [11,12] and *Dendrolimus punctatus* Walker (Lasiocampidae) [13,14], are serious pests of pine plantations in Vietnam and cyclic large-scale outbreaks occur in the dry season every two or three years [15–17]. Severe defoliation reduces volume growth and resin production by up to 12% and 25%, respectively, and repeated defoliation may cause tree mortality [9,13,18,19]. For many years, *Da. axutha* has damaged *P. massoniana* plantations in the Northeast and *De. punctatus* has damaged *P. merkusii* plantations in North Central

Vietnam [9,16–18,20,21]. They also cause severe damage in China [11,12,14]. However, unlike for China, *De. punctatus* causes only minor damage to *P. massoniana* plantations in Vietnam.

The eggs of these species are laid in clusters on needles and the fourth and fifth instars cause the most damage. Although pupation can occur on foliage, the ground and canopies of shrubs, it is concentrated mainly on pine trunks near the ground. Attempts to manage *Da. axutha* and *De. punctatus* have included a range of approaches, including manual, light-trapping, biological and chemical measures [13,19,22–31]. The biological agents *Telenomus dendrolimi*, *T. tetratomus* [28], cytoplasmic polyhedrosid virus [29], *Beauveria bassiana*, *Bacillus thuringiensis* [19,22,30] and the chemical deltamethrin [27] have been applied to try to control these pests.

Integrated pest management (IPM) is widely promoted as the ultimate solution for pest management [32–36] and many models have been deployed [37–39]. Examples are the management of insect pests such as Coleoptera, Lepidoptera, Hemiptera and Orthoptera in *Acacia* plantations [33], *Opisina arenosella* in coconut stands [40] and defoliators in forest plantations in Taiwan [41]. Control thresholds and IPM solutions for *Da. axutha* have been suggested in Vietnam, but so far the work has focused on laboratory trials [24]. Furthermore, IPM of *De. punctatus* has been proposed [13,17], but has not been implemented.

This study aims to determine the survey plan, damage threshold and test integrated management solutions for *Da. axutha* and *De. punctatus* damage in *P. massoniana* and *P. merkusii* plantations in Vietnam using a selection of methods (manual control, light traps, biological agents and synthetic pesticides) based on plantation characteristic and pest density. The findings provide the scientific basis for establishing IPM models, thereby contributing to more effective management of *Da. axutha* and *De. punctatus*. It is anticipated that the research findings will be adopted by the Ministry of Agriculture and Rural Development of Vietnam to disseminate into practice.

## 2. Materials and Methods

### 2.1. Research Approach

The building and testing of IPM models require input data such as the emergence calendar of the pest, plantation characteristics and pest density. Based on the number of pupae, eggs and caterpillars and the average diameter at breast height (Dbh) of the host trees, damage thresholds (DT) are predicted in order to apply management systems. Methods to control the pest population based on the potential damage thresholds are as follows: (1) for <25% leaf area lost, apply manual methods to collect pupae in bark cracks and eggs on the foliage and employ light traps to catch the adults; (2) for 25%–50% leaf area lost, apply manual methods, light traps and biological agents (*Bacillus thuringiensis* var. *kurstaki* + Granulosis virus); and (3) for >50% leaf area lost, apply manual methods, light traps and synthetic pesticides (Deltamethrin, Cypermethrin, or Etofenprox) [19]. The steps undertaken in this study are outlined in Figure 1. Primary model testing (IPM 1 and IPM 2; 2.0 ha per model) was undertaken in 2019 and then six extended models (IPM 1a-c and IPM 2a-c) were employed in 2020.

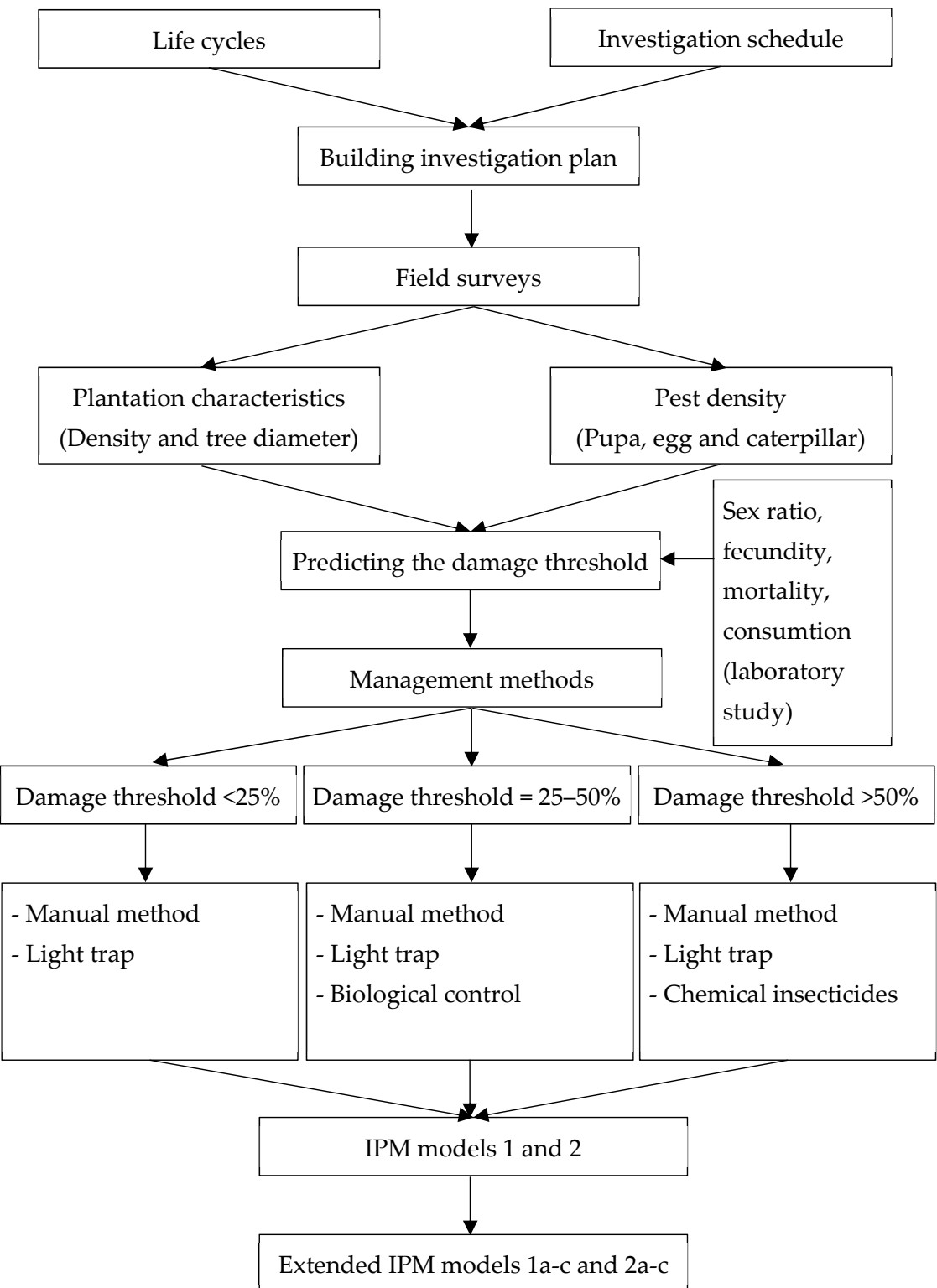

**Figure 1.** Diagram showing the research steps taken to build integrated pest management models for *Dasychira axutha* and *Dendrolimus punctatus*.

*2.2. Laboratory Studies*

The pests were reared in the laboratory in Hanoi to obtain information on fecundity, mortality, fresh weight of pine leaves consumed per caterpillar and the sex ratio. These data are required to help predict the damage threshold. Moths used in experiments were obtained from pupae of *Da. axutha* collected from *P. massoniana* plantations in the Northeast

region and *De. punctatus* from *P. merkusii* plantations in the North Central region of Vietnam. In order to obtain fertilized eggs, ten pairs of newly emerged adults were released into insect cages 0.6 × 0.6 × 1.0 m (length × width × height). Each cage contained fresh branches from 15-year-old *P. massoniana* (for *Da. axutha*) or 37-year-old *P. merkusii* trees (for *De. punctatus*) for laying eggs. Cotton wool inserted into a container of 5% honey solution was placed in each cage to provide food and increase adult longevity. After hatching, 100 caterpillars were placed individually into 9 cm diameter Petri dishes with fresh leaf material, collected from a 15-year-old *P. massoniana* tree in Dong Quan commune, Loc Binh district, Lang Son Province, and a 37-year-old *P. merkusii* tree in Nghi Lam commune, Nghi Loc district, Nghe An Province. They were reared in a laboratory at 30 °C, 85% RH and 12 h light:12 h dark photoperiod. The Petri dishes were inspected every day to replace the leaves and record mortality, weight of pine leaves (g) consumed and the pupa sex ratio.

### 2.3. Field Studies

Field surveillance was carried out from January to December 2018 (three times per month) in order to record the presence and development stages (egg, larva, pupa and adult) in both the Northeast and North Central regions. These data were then used to identify the life cycles and investigation plan of each pest species.

### 2.4. Estimating and Forecasting the Damage Threshold

The average tree diameter at breast height is closely correlated with the fresh leaf biomass. It was calculated according to the following Formula (1) [15] and Formula (2) [42].

$$M_{p.ma} = 1245.83 \times Dbh - 3368.67, \text{ r} = 0.825 \tag{1}$$

$$M_{p.me} = 2110.01 \times Dbh - 20740.12, \text{ r} = 0.95 \tag{2}$$

where $M_{p.ma}$ is the fresh leaf biomass of *P. massoniana* (g); $M_{p.me}$ is the fresh leaf biomass of *P. merkusii* (g); *Dbh* is the diameter at breast height.

Canopy damage was assessed using binoculars on five representative branches in three canopy positions—one branch near the top, two branches in the middle (east–west) and two branches in the lower (south–north) canopy.

The damage threshold (*DT*) was calculated according to the following formula:

$$DT = \frac{F \times L}{M} \times 100 \tag{3}$$

where *DT* is the damage threshold (%); *F* is the theoretical caterpillar population per tree; *L* is the amount of pine leaves (g) consumed by one caterpillar; *M* is the fresh leaf biomass of one tree.

The damage threshold (*DT*) was separated at three levels: (1) light damage, with the leaf area lost <25%; (2) medium damage, with a loss from 25% to 50%; and (3) high damage, with >50% loss.

### 2.5. Building the Integrated Pest Management Models

IPM models 1 and 2 (2.0 ha per plantation in each region) were undertaken in January 2019 (Table 1) to apply suitable management methods (manual, light traps and biological agents) to keep the damage threshold below 25%. Canopy damage was assessed three times per month and the number of pupae, eggs and caterpillars were counted using binoculars in 30 host trees on five representative branches, as described above. At each site, the IPM model and the control were established in the same plantation with similar stocking and site conditions. There was a 30 m buffer between the models. All trees had been used for resin collection from the age of 10 years to 1 month before establishing the field trials. Historically, resin had been collected every five days for nine months a year, excluding the wet season.

**Table 1.** Plantation details where the eight integrated pest management models were established.

| IPM Model | Location | Coordinates | Area (ha) | Age (Year) | Density (tree/ha) | Dbh (cm) | Height (m) | DT (%) |
|---|---|---|---|---|---|---|---|---|
| | IPM models for *Dasychira axutha* in *Pinus massoniana* plantations in Northeast region | | | | | | | |
| IPM 1 | Dong Quan commune, Loc Binh district, Lang Son Province | 21°42′02″ N 106°56′59″ E | 2.0 | 15 | 1040 | 17.1 | 14.2 | 45.8 |
| IPM 1a | Huu Khanh commune, Loc Binh district, Lang Son Province | 21°77′44″ N 106°94′25″ E | 4.6 | 18 | 968 | 18.3 | 15.4 | 42.9 |
| IPM 1b | Cuong Loi commune, Dinh Lap district, Lang Son Province | 21°51′12″ N 107°14′57″ E | 40.3 | 11 | 1056 | 13.7 | 13.5 | 41.5 |
| IPM 1c | Van Son commune, Son Dong district, Bac Giang Province | 21°37′55″ N 106°94′23″ E | 4.1 | 15 | 1023 | 16.8 | 14.0 | 45.5 |
| | IPM models for *Dendrolimus punctatus* in *Pinus merkusii* plantations in North Central region | | | | | | | |
| IPM 2 | Nghi Lam commune, Nghi Loc district, Nghe An Province | 18°50′49″ N 105°33′21″ E | 2.0 | 37 | 435 | 22.2 | 19.2 | 44.3 |
| IPM 2a | Nghi Tien commune, Nghi Loc district, Nghe An Province | 18°86′55″ N 105°67′17″ E | 7.5 | 37 | 443 | 23.2 | 19.6 | 42.8 |
| IPM 2b | Truc Lam commune, Tinh Gia district, Thanh Hoa Province | 19°39′65″ N 105°72′10″ E | 5.4 | 44 | 416 | 28.7 | 19.8 | 43.6 |
| IPM 2c | Ha Linh commune, Ha Trung district, Thanh Hoa Province | 20°01′19″ N 105°79′40″ E | 5.0 | 56 | 408 | 29.5 | 19.7 | 44.0 |

Note: Dbh is diameter at breast height (cm); DT is damage threshold at the time of IPM model construction.

### 2.5.1. IPM Model for *Dasychira axutha* in the Northeast Region

A 2.0 ha area (for IPM 1) was established in January 2019 in a 15-year-old *P. massoniana* plantation in Dong Quan commune, Loc Binh district, Lang Son Province. The manual method was applied in the middle of March and early July 2019. Two people collected and killed the pupae and eggs for one day per ha. Pupae on the trunk were collected by hand. In the canopy, eggs, caterpillars and pupae were located using binoculars. They were then excised using scissors with 5 m handles. Quang's light traps, designed and built by the Vietnamese Academy of Forest Sciences (Hanoi, Vietnam), were used to catch the adults over a week in late March and again in the middle of July 2019. The traps were set up at 1.5 m above the ground, one trap per ha. The traps were placed in the middle of each plot. The battery power light-traps had two light sources, a yellow light to attracts moths from afar and an indigo light to attract nearby moths [16,43]. The living biological agent used to control the neonate caterpillars was *Bacillus thuringiensis* var. *kurstaki* 16.000IU + Granulosis virus $10^8$PIB (Bitadin WP, Nong Sinh Co., Ltd., Hanoi, Vietnam). Bitadin WP is widely used in Vietnam and is recommended by the Ministry of Agriculture and Rural Development for the control of pine tussock moth and masson pine caterpillar [44]. The dosage per ha was 1.2 kg mixed with 6 kg of rice bran powder. Bitadin WP was applied using a two-stroke petrol-powered, high-pressure sprayer (Oshima T3WF-3C-26, Oshima Co., Ltd., Nagasaki, Japan) in early April and 15 days later, as well as in early August 2019 and 15 days later.

### 2.5.2. IPM Model for *Dendrolimus punctatus* in the North Central Region

IPM model 2 (IPM 2) and control model were established in January 2019 in a 37-year-old *P. merkusii* plantation in Nghi Lam commune, Nghi Loc district, Nghe An Province. The manual method was applied in late February and late July 2019. Light traps were

set up in early March and early August 2019. Bitadin WP was applied in mid-April and late August 2019 and it was reapplied after 15 days. The procedures were as described for IPM 1.

2.5.3. Building the Extended Integrated Pest Management Models

From the results of IPM 1 and 2, three extended models (IPM 1a–c) were established in *P. massoniana* plantations in January 2020 in Lang Son and Bac Giang Provinces and three extended models (IPM 2a–c) were established in *P. merkusii* plantations in January 2020 in Nghe An and Thanh Hoa Provinces (Table 1) by forest companies. Staff within the companies were trained to monitor canopy condition, resin collection and application of IPM protocols by researchers from the Forest Protection Research Centre (Hanoi, Vietnam). The methods applied in the six extended IPM models were as applied with IPM 1 and 2, as described earlier.

2.5.4. Data Assessment

Damage was assessed and the number of pupae, eggs and caterpillars were quantified in a random 30 host trees at the time the trial was established; the same trees were analyzed three and seven months later in both IPM and control models. The number of pupae was counted on the trunks, the number of eggs and caterpillars was counted using binoculars for five representative branches in three canopy positions—one branch near the top, two branches in the middle (east–west) and two branches in the lower (south–north) canopy. Damage was assessed using binoculars for the same five representative branches and scored as follows: 0 = no damage (tree healthy); 1 = <25% loss of leaf area; 2 = from 25% to <50% loss; 3 = from 50% to <75% loss; 4 = ≥75% loss (some tree mortality).

Resin production was assessed six times for all trees in April 2020 (IPM 1 and IPM 2) and the extended models were assessed six times in April 2021 based on the process of pine resin exploitation (QTN 29-97) issued together with the Decision No. 2531/QD/NN-KHCN dated 4 October 1997, of the Minister of Agriculture and Rural Development on promulgating the technical process for exploiting dicotyledonous plants [45]. Briefly, minor damage was made to each tree by making a hole into the outer sapwood at a height of 20 cm from the ground and the resin flow was collected using a plastic bag every five days.

*2.6. Data Analysis*

Based on the results of damage classification, the damage incidence $P\%$ was calculated using Equation (4):

$$P\% = \frac{n}{N} \times 100 \tag{4}$$

where $n$ is the number of trees attacked; $N$ is the total number of plants assessed.

The average damage index ($DI$) in each plot was calculated with the following Equation (5):

$$DI = \frac{\sum_0^4 n_i \times v_i}{N} \tag{5}$$

where $n_i$ is the number of trees infected at damage index $i$; $v_i$ is the damage index at level $i$; $N$ is the total number of trees assessed.

Based on the average damage index, the damage severity level was ranked as follows: $DI = 0$ (no damage); $0 < DI \leq 1$ (low damage); $1 < DI \leq 2$ (medium damage); $2 < DI \leq 3$ (high damage); $3 < DI \leq 4$ (very high damage).

The corrected efficacy (%) of each IPM model was calculated according to the following formula by Henderson and Tilton [46]:

$$E = \left(1 - \frac{C_b \times T_a}{C_a \times T_b}\right) \times 100 \tag{6}$$

where $E$ is an inhibitory effect (%); $C_b$ is the damage index of the control at the start of the experiment; $T_b$ is the damage index of the agent treatment at the start of the experiment; $C_a$

is the damage index of the control at the end of the experiment; $T_a$ is the damage index of the agent treatment at the end of the experiment.

## 3. Results

### 3.1. Life Cycles of Dasychira axutha and Dendrolimus punctatus

The caterpillars of *Da. axutha* and *De. punctatus* go through six instars, with five skin molts. *Dasychira axutha* has five generations per year (Table 2) and *De. punctatus* has four generations per year (Table 3).

Female pupa ratios of *Da. axutha* and *De. punctatus* were 42% and 45%, respectively. The average numbers of eggs per female adult of *Da. axutha* and *De. punctatus* were 275 and 297, respectively. The mortality rates of *Da. axutha* and *De. punctatus* were the highest for the pupa and the first instar—48.5%, 51.2% and 51.3%, 50.5%, respectively (Table 4). It seemed that some eggs were unfertilized and some caterpillars were not well adapted to the food and pupated early.

Based on the female pupa ratio, the average numbers of eggs per female adult and mortality rate of each stage, the caterpillar density was calculated with Equation (7) [15]:

$$F = P \times a \times b \times (1 - M) \tag{7}$$

where *F* is the caterpillar density in the next generation; *P* is the density (egg, larva, pupa and adult) in the current generation; *a* is the female pupa ratio; *b* is the average number of eggs per female adult; *M* is the mortality rate of each stage.

The amount of *P. massoniana* and *P. merkusii* leaves consumed by *Da. axutha* and *De. punctatus* in 24 h for all caterpillar stages was 0.65–6.55 g and 0.58–6.18 g per caterpillar, respectively, and was maximal at the fourth instar (Figure 2).

### 3.2. Investigation and Monitoring Plan

Based on Equations (1) and (2), the theoretical fresh leaf biomass of *P. massoniana* and *P. merkusii* at an average Dbh from 10 to <30 cm is 10.3–33.4 kg and 3.5–41.5 kg, respectively (Table 5). Potential damage thresholds can be predicted based on the results for the mortality rate of each development stage, the amount of pine leaves (g) consumed per caterpillar in 24 h and the theoretical fresh leaf biomass for each pine species as a function of the average Dbh.

#### 3.2.1. Investigation Schedule

Based on the life cycle data (Tables 2 and 3) and the number of generations per year, we concluded that the investigation schedules of each generation should be carried out six times: one time in the egg, pupa and adult stages and three times as caterpillars (once in first/second, once in third/fourth and once in fifth/sixth instars) (Table 6).

#### 3.2.2. Investigation Plan

The theoretical damage thresholds for the two pest species in both pine plantations were calculated based on the average number of pupae, eggs and caterpillars per tree for a range of tree diameter size classes (Table 7). The damage threshold was divided into three categories: light damage—the leaf area lost was <25%; medium damage—the leaf area lost was from 25% to 50%; high damage—the leaf area lost was >50%. Methods employed to manage the pests were based on the potential damage thresholds: for light damage, apply manual methods and light traps; for medium damage, apply manual methods, light traps and biological agents; for high damage, apply manual methods, light traps and synthetic pesticides.

**Table 2.** Life cycles and emergence times of *Dasychira axutha* in the Northeast region of Vietnam.

| Generation | Month | | | | | | | | | | | |
|---|---|---|---|---|---|---|---|---|---|---|---|---|
| | **1** | **2** | **3** | **4** | **5** | **6** | **7** | **8** | **9** | **10** | **11** | **12** |
| V | - - | - - - | - 0 0 + + o o | | | | | | | | | |
| I | | | - | - - - | - 0 0 + o | o | | | | | | |
| II | | | | | - | - - - | - 0 0 + | + o o | | | | |
| III | | | | | | | - | - - | - 0 0 + | + o o | | |
| IV | | | | | | | | | - | - - - - | - 0 + o | |
| V | | | | | | | | | | | o | - - |

Note: (-) caterpillar; (0) pupa; (+) moth; (o) egg.

**Table 3.** Life cycles and emergence times of *Dendrolimus punctatus* in the North Central region of Vietnam.

| Generation | Month | | | | | | | | | | | |
|---|---|---|---|---|---|---|---|---|---|---|---|---|
| | 1 | 2 | 3 | 4 | 5 | 6 | 7 | 8 | 9 | 10 | 11 | 12 |
| IV | - - | - 0 0 | 0 + o o | | | | | | | | | |
| I | | | - | - - | - - - 0 0 | + + o o | | | | | | |
| II | | | | | | - | - - - 0 | 0 + o o | | | | |
| III | | | | | | | | - | - - - 0 0 + | o o | | |
| IV | | | | | | | | | | o o - | - - | - - - |

Note: (-) caterpillar; (0) pupa; (+) moth; (o) egg.

**Table 4.** Mortality percentage of each development stage in the laboratory.

| Species | Pupa | Female Adult | Egg | Caterpillar Instar | | | | | |
|---------|------|--------------|-----|-----|-----|-----|-----|-----|-----|
| | | | | 1 | 2 | 3 | 4 | 5 | 6 |
| *Da. axutha* | 48.5 | 8.3 | 18.1 | 51.2 | 26.2 | 11.1 | 17.5 | 15.5 | 12.5 |
| *De. punctatus* | 51.3 | 10.0 | 18.4 | 50.5 | 28.3 | 13.1 | 16.4 | 14.5 | 12.8 |

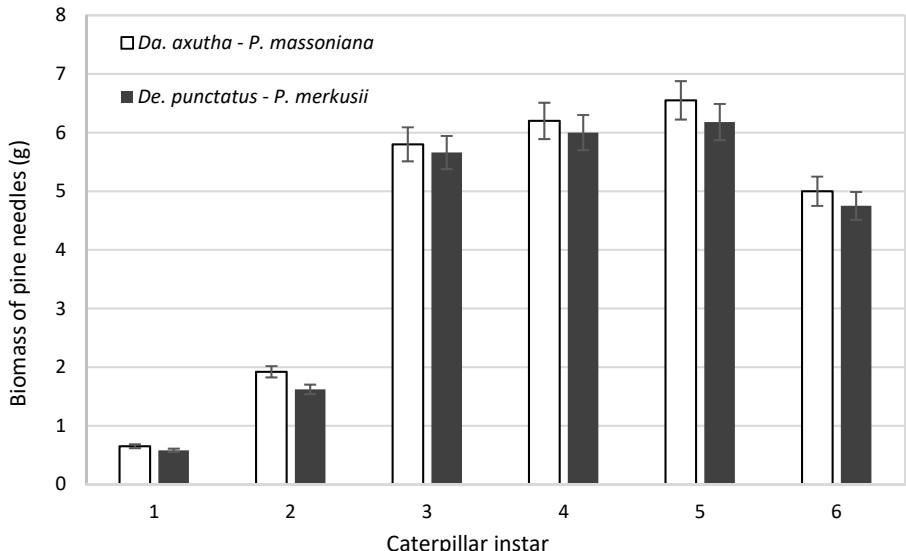

**Figure 2.** Biomass of pine needles (g) consumed per caterpillar in 24 h in the laboratory. The small vertical bars are +/−SE (n = 30).

**Table 5.** Theoretical fresh leaf biomass for pine species by average diameter at breast height.

| Average Dbh (cm) | Amount of Fresh Leaf Biomass (kg) by Host Plant Species | |
|------------------|---------------------------------|---------------------------|
| | *Pinus massoniana* | *Pinus merkusii* |
| 10–<15 | 10.3–15.3 | 3.5–9.9 |
| 15–<20 | 15.3–20.3 | 11.9–20.4 |
| 20–<25 | 22.8–27.8 | 22.5–30.9 |
| 25–<30 | 27.8–33.4 | 33.1–41.5 |

*3.3. Building the IPM Model*

3.3.1. Assessment of IPM Management

The damage incidence and damage index of the IPM models and controls in *P. massoniana* and *P. merkusii* plantations were 35.2%–45.8% and 0.67–0.73, respectively, at the time of model construction. The damage incidence and damage index were greatly reduced in the IPM models at three months and seven months after construction. Meanwhile, the damage index was almost unchanged in the controls and the damage incidence category light increased compared to the initial value. The effect of IPM management of *Da. axutha* damage in *P. massoniana* was 82.4% with the first IPM model (IPM 1) and the effect of IPM 1a, 1b and 1c was 85.9%, 83.4% and 82.4%, respectively. The effect of IPM management of *De. punctatus* damage in *P. merkusii* with the first IPM model (IPM 2) was 77.8% and the other models (IPM 2a, 2b, 2c) had efficacies of 80.4%, 79.5% and 79.2%, respectively (Table 8).

**Table 6.** Determined schedules for *Dasychira axutha* in Northeast region and *Dendrolimus punctatus* in North Central region.

| Generation | Investigation Schedule of Development Stages | | | |
|---|---|---|---|---|
| | Egg | Caterpillar | Pupa | Adult |
| *Dasychira axutha* in Northeast region | | | | |
| 1 | 20/3–10/4 | 01/4–30/4 | 05/3–20/3 | 15/3–25/3 |
| 2 | 10/5–30/5 | 25/5–25/6 | 05/5–20/5 | 10/5–20/5 |
| 3 | 20/7–10/8 | 01/8–30/9 | 05/7–20/7 | 10/7–20/7 |
| 4 | 20/9–10/10 | 05/10–25/11 | 15/9–30/9 | 25/9–05/10 |
| 5 | 20/11–10/12 | 10/12–30/01 | 20/11–30/11 | 25/11–05/12 |
| *Dendrolimus punctatus* in North Central region | | | | |
| 1 | 10/3–30/3 | 25/3–15/5 | 15/2–05/3 | 05/3–15/3 |
| 2 | 05/6–20/6 | 10/6–20/7 | 10/5–25/5 | 20/5–10/6 |
| 3 | 01/8–20/8 | 15/8–20/9 | 20/7–05/8 | 01/8–10/8 |
| 4 | 15/10–30/10 | 05/11–15/01 | 20/9–10/10 | 01/10–10/10 |

**Table 7.** Average number of pupae, eggs and caterpillars per tree and the damage threshold according to the average diameter at breast height.

| Average Diameter at Breast Height (cm) | Number of Pupae and DT | | Number of Eggs and DT | | Number of Caterpillars and DT | | | | | |
|---|---|---|---|---|---|---|---|---|---|---|
| | | | | | Age 1–2 | | Age 3–4 | | Age 5–6 | |
| | 50% | 25% | 50% | 25% | 50% | 25% | 50% | 25% | 50% | 25% |
| *Dasychira axutha* | | | | | | | | | | |
| 10–<15 | 2.2 | 1.1 | 122.8 | 61.6 | 100.8 | 50.4 | 89.5 | 44.7 | 50.0 | 25.0 |
| 15–<20 | 3.4 | 1.7 | 189.2 | 95.1 | 155.4 | 77.7 | 138.0 | 69.0 | 77.2 | 38.6 |
| 20–<25 | 4.8 | 2.4 | 263.3 | 132.8 | 216.1 | 108.1 | 192.0 | 96.0 | 107.3 | 53.7 |
| 25–<30 | 6.2 | 3.1 | 338.4 | 168.5 | 276.9 | 138.4 | 245.9 | 122.9 | 137.5 | 68.7 |
| *Dendrolimus punctatus* | | | | | | | | | | |
| 10–<15 | 0.6 | 0.3 | 32.7 | 15.8 | 25.7 | 12.9 | 23.6 | 11.3 | 12.4 | 6.2 |
| 15–<20 | 2.8 | 1.4 | 151.9 | 76.2 | 124.7 | 62.3 | 109.5 | 54.7 | 60.3 | 30.1 |
| 20–<25 | 5.2 | 2.6 | 286.8 | 143.3 | 234.6 | 1173 | 206.0 | 103.0 | 113.4 | 56.7 |
| 25–<30 | 7.7 | 3.8 | 421.5 | 209.8 | 344.6 | 172.3 | 302.5 | 151.3 | 166.5 | 83.3 |

Note: DT is damage threshold; 25% and 50% are percentages of leaf area lost.

### 3.3.2. Assessment of Resin Productivity

The decrease in damage incidence and damage index led to an increase in resin production of *P. massoniana* and *P. merkusii* plantations by 20.9%–22.7%, compared to the control (Table 9).

**Table 8.** Effect of the implementation of integrated pest management on damage incidence (P%) and damage index (DI).

| Model | Before IPM Model Implementation | | 3 Months after IPM Model Implementation | | 7 Months after IPM Model Implementation | | Efficacy (%) |
|---|---|---|---|---|---|---|---|
| | P% | DI | P% | DI | P% | DI | |
| IPM 1 | 45.8 | 0.69 | 14.2 | 0.23 | 6.4 | 0.12 | 82.4 |
| Control | 40.6 | 0.73 | 46.3 | 0.76 | 50.7 | 0.72 | |
| IPM 1a | 42.9 | 0.71 | 15.1 | 0.24 | 8.6 | 0.11 | 85.9 |
| Control | 36.3 | 0.62 | 45.8 | 0.71 | 54.4 | 0.68 | |
| IPM 1b | 41.5 | 0.53 | 14.9 | 0.25 | 8.3 | 0.10 | 83.4 |
| Control | 39.9 | 0.44 | 47.7 | 0.51 | 46.2 | 0.50 | |
| IPM 1c | 45.5 | 0.75 | 9.5 | 0.21 | 9.1 | 0.16 | 82.4 |
| Control | 44.6 | 0.71 | 50.4 | 0.80 | 48.2 | 0.86 | |
| IPM 2 | 44.3 | 0.73 | 14.8 | 0.29 | 10.3 | 0.15 | 77.8 |
| Control | 41.1 | 0.67 | 51.2 | 0.82 | 47.6 | 0.62 | |
| IPM 2a | 42.8 | 0.43 | 14.9 | 0.26 | 8.1 | 0.11 | 80.4 |
| Control | 35.2 | 0.39 | 49.2 | 0.52 | 45.4 | 0.51 | |
| IPM 2b | 43.6 | 0.53 | 15.5 | 0.30 | 8.6 | 0.13 | 79.5 |
| Control | 42.0 | 0.51 | 49.0 | 0.59 | 45.9 | 0.61 | |
| IPM 2c | 44.0 | 0.77 | 9.9 | 0.20 | 9.6 | 0.16 | 79.2 |
| Control | 43.3 | 0.75 | 50.6 | 0.88 | 48.8 | 0.75 | |

**Table 9.** The effect of the integrated pest management models on resin productivity.

| Model | Resin Productivity (kg/ha/month) * | | Increase over the Control (%) |
|---|---|---|---|
| | IPM Model | Control | |
| IPM 1 | 163.3 | 135.0 | 21.0 |
| IPM 1a | 168.3 | 138.3 | 21.7 |
| IPM 1b | 154.2 | 126.7 | 21.7 |
| IPM 1c | 164.3 | 134.5 | 22.2 |
| IPM 2 | 140.0 | 115.8 | 20.9 |
| IPM 2a | 93.8 | 76.5 | 22.7 |
| IPM 2b | 94.3 | 77.4 | 21.7 |
| IPM 2c | 93.3 | 77.2 | 20.9 |

Note: * Each value is the mean of all trees (see Table 1) sampled 5 times in one month.

## 4. Discussion

*Pinus massoniana* and *P. merkusii* plantations are environmental protection and production forests of great economic and environmental value in Vietnam [8,9]. Because of the extent and frequency of damage incurred by *Dasychira axutha* and *Dendrolimus punctatus* [13,16,18,20], there is a pressing need for approved pest management protocols. This study is the first to provide the damage threshold, survey schedule and integrated management solutions for *Da. axutha* and *De. punctatus* in *Pinus* plantations in Vietnam.

Data obtained from laboratory experiments and field surveys were used in the construction of investigation schedules and to identify damage thresholds. Life cycle data are critical for determining IPM protocols [39]. The life cycle data we used appeared to be robust, as they were similar to the data obtained in previous studies on the total number of days to complete the life cycle (for *Da. axutha*, 65–78 days, [20] and, for *De. punctatus*, 82–98 days [21]). The average mortality of *Da. axutha* and *De. punctatus* caterpillars in our study in the laboratory was 22.3%–22.6%. Caterpillars of this species had a mortality of 13.6%–54.0% on one-year-old *P. massoniana* trees in China [47,48]. Our study determined

that the eating capacity of *Da. axutha* and *De. punctatus* caterpillars was 4.35 g and 4.13 g of fresh leaves per caterpillar over a 24 h period, respectively. This provides an important basis for determining the control threshold for each pest and each host plant species. Comparable values obtained in laboratory studies in Guizhou (China) were 4.12 g [49] and 4.05–4.25 g [24,49], respectively. Leaf age and larval host species can affect larval performance and adult reproductive fitness [48].

Our study determined the survey schedule for *Da. axutha* and *De. punctatus* damage in *P. massoniana* and *P. merkusii* plantations in the Northeast and North Central regions, respectively. Forecasting is important for pest management. Bayes' discriminant analysis was used to forecast the peak occurrence of *De. punctatus* with an accuracy of 85.7%–97.2% in China [50,51]. So far, this approach has not been applied to forests in Vietnam. Previous studies in Vietnam only recommended monitoring litter depth [17,20,21]. The lack of survey schedules caused difficulties for forecasting activities and was costly in terms of time and money. Forecasting is of great interest and researchers in China have identified and calibrated survey parameters to increase the accuracy of forecast results [52]. Furthermore, a PCA-RF detection model was shown to be sufficiently robust for forest pest detection [53].

Using formulas [15,42], the amount of fresh leaf biomass was determined for each diameter class (Table 6) and this was combined with the feeding capacity of the caterpillars to determine damage thresholds and to build the IPM models. Previously, the feeding capacity of *Da. axutha* was tested [24], but the authors did not identify the damage thresholds. For each damage threshold, appropriate treatment solutions were applied to achieve the best effect.

Ultra-low volume spays have been used to control *Da. axutha* caterpillars (from the 2nd to the 4th instars) with 85.1%–94.7% control [23]. Sex pheromone-baited traps were used to catch and kill adult males of *De. punctatus* [54]. Light traps were used to catch adult *De. punctatus* [13] and *Da. axutha* [19,31]. Biological control has also been used with some lepidopterous pine pests in China. Scelionid egg parasitoids, *Telenomus dendrolimi* and *T. tetratomus*, have been used against *De. punctatus* and *Da. axutha* [28]. Tiny *Trichogramma dendrolimi* egg parasitoid wasps, containing a cytoplasmic polyhedrosid virus, were used to control *De. punctatus* in China [29]. The parasitic fungus *Beauveria bassiana* was used to control caterpillars of *De. punctatus* [25,26,30]. A mixture of *B. bassiana*, *Bacillus thuringiensis* and *T. dendrolimi* virus was used to control *Da. axutha* and *De. punctatus* [19,22,30]. Additionally, *B. bassiana* was mixed with a pyrethoid (deltamethrin) insecticide resulting in high mortality of *De. punctatus* in pine forests in China [27].

Unlike in many countries [32,36,39], Vietnam has been slow to adopt IPM. For example, IPM has been used for controlling insect pests in *Acacia* plantations [33] and leaf-cutting ants damaging *Pinus* and *Eucalyptus* plantations in Brazil [55,56]. The deployment of IPM for *Da. axutha* and *De. punctatus* in Vietnam reduced canopy damage by 82.4%–85.9% and 77.8%–80.4%, respectively, and increased resin yield by more than 20% compared to the control. The increase in resin yield was likely due to the greater availability of photosynthate for oleoresin synthesis in trees with reduced canopy damage. This is consistent with studies that have shown that resin production can be influenced by abiotic and biotic stress [18,19,57,58]. Previous research works have shown that severely damaged *P. massoniana* and *P. merkusii* forests can experience a 15%–25% reduction in resin yield compared to undamaged or lightly damaged forests [19,31]. Pine resins provide raw materials for the pharmaceutical, cosmetic, food and chemical industries [59]. The collection of pine resin provides an important source of cash for the local communities who live among the pine forests in Vietnam [10]. Thus, reducing the damage from pine caterpillars can help to maintain resin production and stabilize growers' livelihoods across the years.

Management measures for the two pest species, especially for *De. punctatus*, have been studied and applied for many years in Vietnam [13,15,20,21,24]. However, these measures have not been very effective and infestations frequently occur. The IPM models established in our research study provide an opportunity to better manage these pests in the future. The damage index in *P. massoniana* and *P. merkusii* plantations before implementing the IPM

was 0.53–0.75 and 0.43–0.77, respectively; after IPM implementation, it was 0.10–0.16 and 0.11–0.16, respectively. The models were applied twice in one year. Because the damage incidence of IPM 1c and 2c was less than 10% after 3 months of treatment, it was only necessary to use the manual method and light traps the second time.

Forest pests are becoming more prevalent in Vietnam [60] and pesticides are becoming more widely used to control pests in forest plantation species. Examples of pesticide use include *Da. axutha* and *De. punctatus* in *Pinus* plantations [19,24], *Hypsipyla robusta* in *Chukrasia tabularis* plantations [61] and *Batocera lineolata* in *Eucalyptus* plantations [62]. However, the government in Vietnam is concerned about pesticide residues entering the human food chain and is encouraging a move towards more environmentally friendly pest control measures that meet international certification protocols [63,64]. Our study demonstrates that, if IPM models are applied early enough, the deployment of insecticides can be prevented. Forest companies in Vietnam are very receptive to adopting IPM and reducing the use of pesticides. The forest companies we worked with confirmed the feasibility of the extended IPM models in terms of effectiveness in pest control, operational efficiency and cost. To extend IPM uptake, an operations' manual will be produced for the forest sector through the State Forestry Administration. Further research is planned with the aim to assess other biological agents, such as *Beauveria bassiana* [25–27], *Bacillus thuringiensis* [19] and baculovirus (*B. thuringiensis kurstaki* (Btk) strain ABTS-351) [65]. To reduce manual labor, we will also explore whether trunk trap nets are effective [66].

## 5. Conclusions

The IPM models we developed for *Da. axutha* and *De. punctatus* in *P. massoniana* and *P. merkusii* plantations were effective in reducing damage without the need for insecticides. Moreover, they were accepted as workable by 15 forest companies; therefore, they could be attractive more broadly in the forest sector. The IPM protocols can now be disseminated to all producers engaged in *Pinus* production in Vietnam. It is anticipated that full deployment would result in better outcomes for tree growth and rural economies in Vietnam.

**Author Contributions:** Conceptualization, D.N.Q.; setting trial, D.N.Q. and P.Q.T.; investigation and sampling, D.N.Q., L.V.B., N.Q.T. and V.D.N.; rearing and implementation, D.N.Q., L.V.B., N.Q.T., N.H.T., N.M.C. and V.D.N.; analysis, D.N.Q. and N.M.C.; methodology, D.N.Q., B.D., N.M.C., L.V.B. and P.Q.T.; validation, D.N.Q. and P.Q.T.; visualization, D.N.Q. and B.D.; writing—original draft, D.N.Q. and N.M.C.; writing—review and editing, B.D., D.N.Q. and N.M.C. All authors have read and agreed to the published version of the manuscript.

**Funding:** This research study was supported by the Ministry of Agriculture and Rural Development of Vietnam, under the project "Study on measures to control caterpillars feeding on *Pinus merkusii* and *Pinus massoniana* in North and North Central Vietnam, code: 3046/QĐ-BNN-KHCN".

**Institutional Review Board Statement:** Not applicable.

**Informed Consent Statement:** Not applicable.

**Data Availability Statement:** Not applicable.

**Acknowledgments:** The authors would like to thank Loc Binh Forest Company, Dinh Lap Forest Company, Lang Son province, Son Dong Forest Company, Bac Giang province, Management Board for Nghi Loc Protection Forest, Nghe An province, Ha Trung Forest Company and Management Board for Tinh Gia Protection Forest, Thanh Hoa province, for their support in setting up the models and collecting data, as well as Gondess Pty Ltd. for travel support to BD.

**Conflicts of Interest:** The authors declare no conflict of interest. The funders had no role in the design of the study; in the collection, analyses, or interpretation of data; in the writing of the manuscript, or in the decision to publish the results.

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
