# Peer review of "Management of Needle-Eating Caterpillars Associated with Pinus massoniana and P. merkusii in Vietnam"

_forests, doi:10.3390/f12111610_

Round 1

Reviewer 1 Report

please see the attached file 

Author Response

Dear Reviewer,

Yours sincerely,

Dao Ngoc Quang

Reviewer 2 Report

The article was well written, and have important information not just for the management of those caterpillars on the forests of Vietnam, but for the management of caterpillars that are pests in forestry in general. The kind of information that is included in this manuscript is very important to forest pests management, but is rare to find on the literature.  I pointed out some small issues through this review (attached).

Author Response

(The authors gave the same response as above.)
